# Dietary nitrate supplementation prevents radiotherapy-induced xerostomia

Xiaoyu Feng[1,2,3†], Zhifang Wu[1,2†], Junji Xu[1,2], Yipu Xu[1,2], Bin Zhao[1,2], Baoxing Pang[1,2], Xingmin Qu[1,2], Liang Hu[1,2], Lei Hu[1,2], Zhipeng Fan[1,2], Luyuan Jin[1,2], Dengsheng Xia[1,2], Shimin Chang[1,2], Jingsong Wang[1,4], Chunmei Zhang[1,2], Songlin Wang[1,2,4*]

[1]Beijing Laboratory of Oral Health, Capital Medical University, Beijing, China; [2]Salivary Gland Disease Center and Molecular Laboratory for Gene Therapy & Tooth Regeneration, Beijing Key Laboratory of Tooth Regeneration and Function Reconstruction, School of Stomatology, Capital Medical University, Beijing, China; [3]Department of Pediatric Dentistry, Capital Medical University School of Stomatology, Beijing, China; [4]Biochemistry and Molecular Biology, School of Basic Medical Sciences, Beijing, China

**\*For correspondence:**
slwang@ccmu.edu.cn

[†]These authors contributed equally to this work

**Abstract** Management of salivary gland hypofunction caused by irradiation (IR) therapy for head and neck cancer remains lack of effective treatments. Salivary glands, especially the parotid gland, actively uptake dietary nitrate and secrete it into saliva. Here, we investigated the effect of dietary nitrate on the prevention and treatment of IR-induced parotid gland hypofunction in miniature pigs, and elucidated the underlying mechanism in human parotid gland cells. We found that nitrate administration prevented IR-induced parotid gland damage in a dose-dependent manner, by maintaining the function of irradiated parotid gland tissue. Nitrate could increase sialin expression, a nitrate transporter expressed in the parotid gland, making the nitrate-sialin feedback loop that facilitates nitrate influx into cells for maintaining cell proliferation and inhibiting apoptosis. Furthermore, nitrate enhanced cell proliferation via the epidermal growth factor receptor (EGFR)–protein kinase B (AKT)–mitogen-activated protein kinase (MAPK) signaling pathway in irradiated parotid gland tissue. Collectively, nitrate effectively prevented IR-induced xerostomia via the EGFR–AKT–MAPK signaling pathway. Dietary nitrate supplementation may provide a novel, safe, and effective way to resolve IR-induced xerostomia.

## Introduction

Head and neck cancer (HNC) is the sixth most common malignancy worldwide (*Lindsey et al., 2012*). Irradiation (IR) is an important treatment approach for HNC. Salivary glands, which are often included in the IR field, are highly radiosensitive, and IR often results in salivary gland hypofunction. IR destroys salivary gland acinar cells, the sole site of fluid transport in the gland parenchyma, resulting in xerostomia, which severely impacts quality of life for affected patients (*Bhide et al., 2009*; *Davies and Thompson, 2015*; *Mercadante et al., 2017*).

Acinar cells in salivary glands include both serous and mucous cell types. Serous acinar cells are much more radiosensitive than mucous acinar cells. Notably, the parotid gland consists solely of serous acinar cells, which makes the parotid gland more radiosensitive than the submandibular and sublingual gland (*Grundmann et al., 2009*). As such, the parotid gland provides an excellent model organ to study IR-induced salivary gland hypofunction. Many studies have attempted to treat salivary hypofunction using gene transfer, stem cell transplantation, or salivary gland regeneration. These studies have elucidated the mechanisms involved in xerostomia, and allowed for the development of

**eLife digest** Head and neck cancers are commonly treated using radiotherapy, where a beam of high-energy radiation is targeted at the tumour. This often severely damages the surrounding salivary glands, leading to chronic dry mouth and impairing a patient's sense of taste, nutrient intake, speech and immune system. Despite this significant impact on quality of life, there is no effective treatment yet for this side effect.

In the body, salivary glands are one of the primary users of a compound known as nitrate, which is commonly found in the diet. In the glands, it is ushered into cells thanks to a protein known as sialin. The nutrient supports the activity and maintenance of the glands, before it is released in the saliva. Feng, Wu et al. therefore decided to test whether nitrate could offer protection during neck and head radiotherapy.

The experiments used miniature pigs, which have similar salivary glands to humans. The animals that received sodium nitrate before and after exposure to radiation preserved up to 85% of their saliva production. By comparison, without any additional nitrate, saliva production fell to 20% of pre-radiation levels.

To understand how this protective effect emerged, Feng, Wu et al. added nitrate to cells from a human salivary gland known as the parotid. This led to the cells producing more sialin, creating a feedback loop which increases the amount of nitrate in the salivary glands. Further examination then showed that the compound promotes growth of cells and reduce their death. These findings therefore suggest that clinical studies may be worthwhile to test if nitrate could be used to prevent dry mouth in head and neck cancer patients who undergo radiotherapy.

new treatments for xerostomia (*Vissink et al., 2010*; *Vissink et al., 2015*). However, most of these treatment modalities are therapeutic and thus beneficial for patients with established xerostomia (i.e., when salivary glands are severely damaged) and have not been adequately resolved IR-induced xerostomia (*Vissink et al., 2010*). In addition, there have been no studies on the prevention of IR-induced xerostomia. New approaches for the prevention of IR-induced xerostomia are urgently required to maintain the tissue architecture of the parotid gland even after IR treatment.

Inorganic nitrate is a component of the human diet, and is found in vegetables, fruits, and drinking water. At least 25 % of circulating nitrate is actively taken up by salivary glands (via sialin) and secreted into saliva, such that the concentration of nitrate in saliva is approximately tenfold greater than that in blood (*Omar et al., 2016*). Nitrate has been shown to offer protection against diseases such as obesity, diabetes mellitus, and heart disease (*Gilchrist et al., 2014*; *Khambata et al., 2017*; *Kina-Tanada et al., 2017*; *Lundberg et al., 2018*; *Ma et al., 2018*; *McNally et al., 2016*; *Omar et al., 2016*). Sialin (*Slc17a5*) is a transmembrane protein expressed most abundantly in the acinar cells of the parotid gland (*Qin et al., 2012*). Sialin transports nitrate and other essential cellular substances such as glutamate and aspartate, and contributes to the maintenance of acinar cells physiological function (*Qin et al., 2012*). It is unclear why sialin is most strongly expressed in the parotid gland and facilitates nitrate influx into the gland. The interaction of sialin and nitrate is unique to the parotid gland and not found in other organs. It is highly likely that the interaction between nitrate and sialin contributes to the maintenance of parotid gland homeostasis.

The parotid glands of mice and rats are not similar to those of humans (*Wang et al., 1998*). We previously demonstrated that the parotid glands of miniature pigs have similar anatomical and physiological characteristics to those of human parotid glands. Then we established miniature pig model of IR-induced parotid gland hypofunction using single-dose or fractionated IR (*Guo et al., 2014*; *Li et al., 2005*). In this study, we investigated the preventive and therapeutic effects of dietary nitrate supplementation on IR-induced salivary gland hypofunction using an established miniature pig model, and characterized the mechanism underlying these effects.

## Results

### Nitrate prevents but does not correct IR-induced parotid gland hypofunction in vivo

First, we investigated whether inorganic nitrate could prevent or treat IR-induced xerostomia. Miniature pigs were administered a single dose of 20 Gy (biological dose equal to 60 Gy) (*Figure 1a and d* and *Figure 1—figure supplement 1a, b*), and the effects of preventive and therapeutic nitrate administration were evaluated (*Figure 1a–f*). The IR control group exhibited a sharp decrease in mean salivary flow rate (SFR) (*Figure 1b and e*). In the preventive nitrate group, the mean SFR was significantly decreased at 1 -month post-IR, but recovered to nearly normal levels at 4 months post-IR. Long-term observation (2 years post-IR) showed that SFR was at 80 % of pre-IR levels (*Figure 1b*). However, the mean SFR in the therapeutic nitrate group did not show significant recovery and remained similar to that in the IR control group (*Figure 1e*). The mean salivary concentrations of amylase, sodium, and chloride in saliva were significantly higher in the preventive nitrate group than in the IR control group (*Figure 1—figure supplement 1c*). In contrast, the mean potassium concentration was significantly lower following nitrate administration (*Figure 1—figure supplement 1c*).

The mean change in local blood flow rate in irradiated parotid glands displayed a pattern similar to that of the mean SFR. Local blood flow rate was stable in the preventive nitrate group post-IR, but decreased sharply with time in the therapeutic and IR control groups (*Figure 1c and f*). Four months post-IR, SFR reduction and parotid gland tissue degeneration had stabilized (the latter was determined using terminal deoxynucleotidyl transferase dUTP nick-end labeling in situ assay) (*Figure 1—figure supplement 1d*). Histological analyses showed that far more acinar cells remained and less fibrosis was present at 4 months post-IR in the preventive nitrate group compared to the therapeutic and control groups (*Figure 1g*).

### Nitrate protects parotid gland tissue against IR damage in a dose-dependent manner

Next, we studied the dose-dependent effects of nitrate on IR-induced salivary gland hypofunction using a fractionated IR model that mimics clinical conditions (*Figure 2a*). Exogenously administered nitrate was well absorbed, as evidenced by significantly higher salivary and serum nitrate concentrations in the nitrate groups compared with those in the control groups. This difference was more pronounced at higher nitrate doses (*Figure 2—figure supplement 1a*, b).

The mean SFR gradually decreased with time after unilateral parotid gland fractionated IR ('IR control' in *Figure 2b*). Four months post-IR, the mean SFR in the IR control group was approximately 20 % of that observed pre-IR (*Figure 2b'*). Conversely, mean SFRs were protected from IR-induced changes in a dose-dependent manner (*Figure 2b*). The highest mean SFR was observed in the group that received 2 mmol/kg·day nitrate, with a mean SFR of nearly 85 % of the pre-IR SFR. The mean SFRs were 65%, 50%, and 30 % of those observed before IR in response to nitrate doses of 1, 0.5, and 0.25 mmol/kg·day, respectively (*Figure 2b'*).

The mean changes in local blood flow rate showed a pattern similar to those of the mean SFRs. The mean local blood flow rate was highest for the group that received the highest nitrate dose, and this effect decreased in a dose-dependent manner (*Figure 2c*). In addition, nitrate reduced irradiated parotid gland weight in a dose-dependent manner, and there was no significant difference between parotid gland weights in the highest-dose group (2 mmol/kg·day) and the sham group (*Figure 2d*).

Several salivary constituents (amylase, calcium, and potassium) were significantly altered following nitrate administration, particularly at 4 months post-IR (*Figure 2—figure supplement 1c,e*).

At 4 months post-IR, hematoxylin and eosin (H&E) staining, and Masson staining, showed that most acinar cells were lost in parotid glands in the IR control group. Moreover, the ductal system was irregularly expanded, bent, or blocked, and exhibited extensive fibrosis, which was typical of IR-induced salivary gland damage (*Figure 2e*). However, parotid gland morphology was generally maintained in the nitrate groups in a dose-dependent manner. In the highest-dose group (2 mmol/kg·day), the morphology was nearly identical to that of the sham group (*Figure 2e*). These results demonstrated that nitrate contributed to the preservation of function and morphology in irradiated parotid glands in a dose-dependent manner.

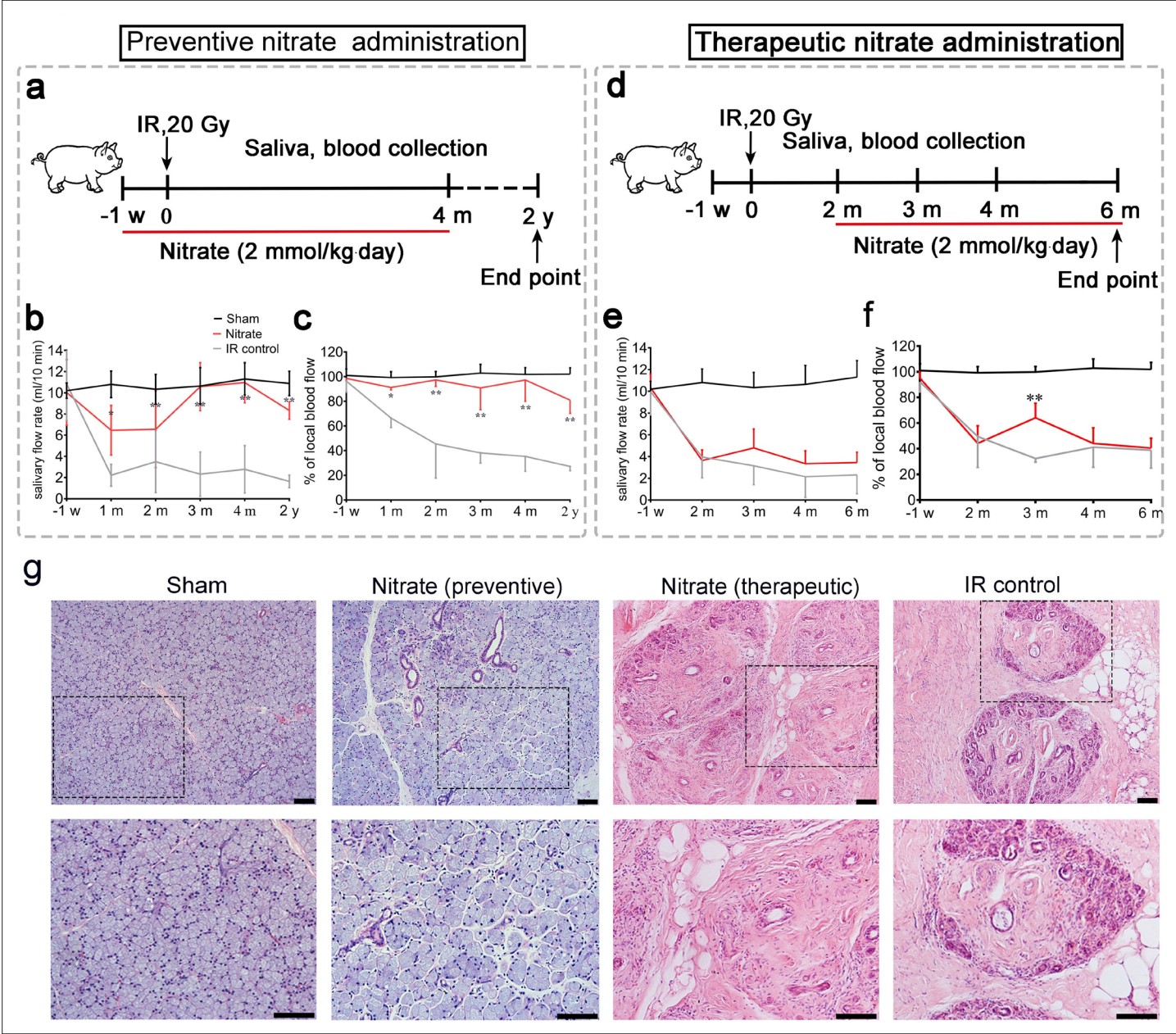

**Figure 1.** Preventive and therapeutic effects of nitrate administration on the parotid gland in miniature pigs subjected to single-dose irradiation (IR). (**a–c**) Preventive nitrate administration to miniature pigs. (**a**) Schematic of experimental design. (**b**) Nitrate administration maintained salivary flow rate (SFR) (ml/10 min) of the parotid gland in the sham and preventive nitrate groups, but not in the IR control group (n=4 animals per group); the sham group received 0 Gy, and the nitrate and IR control groups received 20 Gy. (**c**) Change in local blood flow rate (%) in the parotid gland in the sham, preventive nitrate, and IR control groups (n=4 animals per group). (**d–f**) Therapeutic nitrate administration to miniature pigs. (**d**) Schematic of experimental design. (**e**) SFR (ml/10 min) of the parotid gland in the sham, therapeutic nitrate, and IR control groups (n=4 animals per group); the sham group received 0 Gy, and the nitrate and IR control groups received 20 Gy. (**f**) Change in local blood flow rate (%) in the parotid gland in the sham, therapeutic nitrate, and IR control groups (n=4 per group). Data are expressed as the mean ± standard error of the mean (SEM). *p<0.05, **p<0.01, nitrate group versus IR control group at different time points. The sham group served as baseline for reference (**b**), (**c**), (**e**), and (**f**). (**g**) Hematoxylin and eosin staining of parotid gland sections in the sham, preventive nitrate, therapeutic nitrate, and IR control groups at 4 months post-IR. Scale bar: 100 μm.

The online version of this article includes the following figure supplement(s) for figure 1:

**Figure supplement 1.** Parotid gland irradiation (IR) range diagrams, blood vessel distribution, salivary chemistry, and tissue apoptosis at 4 months after IR.

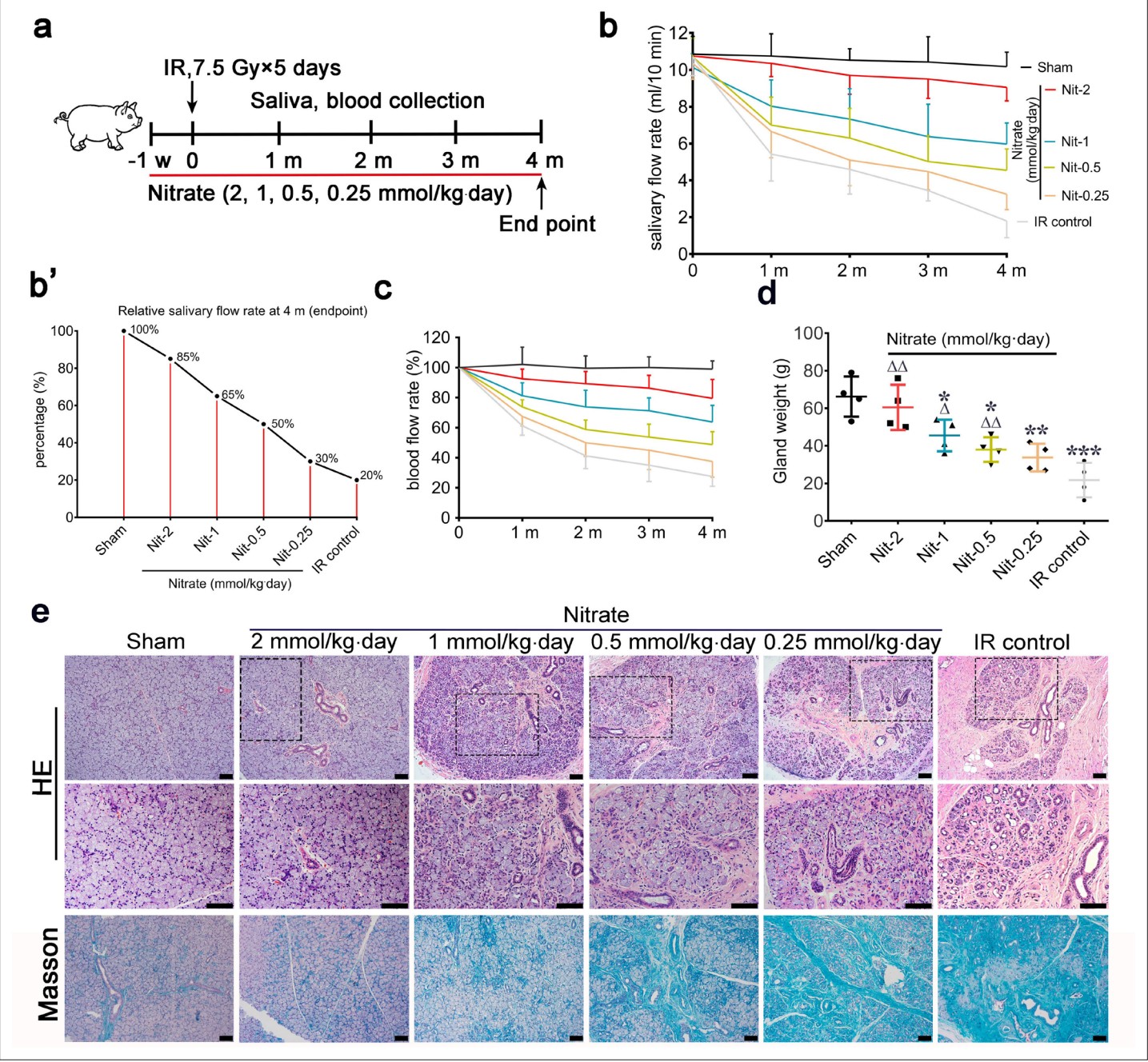

**Figure 2.** Effects of different preventive nitrate doses on the parotid gland in miniature pigs subjected to fractionated irradiation (IR). (**a**) Schematic of the experimental design for preventive nitrate administration of four different doses of nitrate (0.25, 0.5, 1, or 2 mmol/kg·day) to miniature pigs. (**b**) Salivary flow rate (SFR) of the parotid gland (ml/10 min) in the sham group, the four nitrate groups, and the IR control group (n=4 animals per group; data are expressed as mean ± SEM); the sham group received 0 Gy, and the nitrate groups and the IR control group received 7.5 Gy for five consecutive days. (**b'**) SFR at 4 months post-IR compared with pre-IR SFR in the sham group, the nitrate groups, and the IR control group. (**c**) Change in local blood flow rate (%) in the parotid gland in the sham group, the nitrate groups, and the IR control group (n=4 animals per group; data are expressed as the mean ± SEM). Treatment groups correspond to those shown in (**b**). (**d**) Weight of the parotid gland in the sham group, the nitrate groups, and the IR control group (n=4 animals per group; data are expressed as the mean ± SEM, *p<0.05, **p<0.01, ***p<0.005 (each nitrate group compared with the sham group); Δ<0.05, ΔΔ<0.01 (each nitrate group compared with the IR control group). (**e**) Hematoxylin and eosin, and Masson trichrome staining of parotid gland sections in the sham group, the nitrate groups, and the IR control group (n=4 animals per group). Scale bar: 100 μm. SEM, standard error of the mean.

The online version of this article includes the following figure supplement(s) for figure 2:

**Figure supplement 1.** Nitrate concentrations in serum (**a**) and parotid saliva (**b**), and changes in salivary constituents at 4 months after fractionated irradiation (IR).

## Nitrate increases cell proliferation and microvessel density

Nitrate administration significantly increased the proliferation of acinar and ductal cells as determined by Ki67 expression (*Figure 3a*). Furthermore, microvessel density (MVD; measured by CD31 expression) was significantly increased following nitrate administration, whereas the IR control group showed marked MVD loss (*Figure 3b*). Aquaporin 5 (AQP5), a water channel essential for saliva secretion, was highly expressed in the acini of the nitrate group (*Figure 3c*), further suggesting that the secretory function of the remaining parotid gland tissue was preserved in response to nitrate administration. Surprisingly, sialin expression was increased after nitrate administration (*Figure 3d*), suggesting there is a close interaction between sialin and nitrate. Moreover, the results of nitrate administration to human parotid gland cells (hPGCs) in vitro confirmed that nitrate had a dose-dependent positive effect on sialin expression (*Figure 3e and f*, *Figure 3—source data 1* and *Figure 3—source data 2*).

Overall, these findings suggested that preventive nitrate administration protected the parotid gland against IR damage via nitrate-sialin interaction. Because sialin is known to facilitate the influx of nitrate and other essential cellular substances, we next investigated the roles of sialin in parotid gland cells.

## Nitrate is necessary to maintain cell survival, promote hPGPCs proliferation, and inhibit apoptosis

First, we observed nitrate administration on cell proliferation and apoptosis to hPGCs in vitro under post-IR (5 Gy) culturing conditions or physiological conditions (*Figure 4a* and *Figure 4—figure supplement 1a*). We found that nitrate administration prior to IR (*Figure 4a*) significantly increased cell proliferation (*Figure 4b and c*), but little effect on cell apoptosis (*Figure 4d*). Moreover, nitrate administration to hPGCs in vitro resulted in a dose-dependent increase in sialin expression (*Figure 4e* and *Figure 4—source data 1*). Consistently, nitrate administration to hPGCs under physiological conditions also exhibited increased cell proliferation and elevated sialin expression in a dose-dependent manner (*Figure 3f* and *Figure 4—figure supplement 1a,d*). We knocked down (with short interfering RNA; siRNA) or overexpressed sialin (*Slc17a5*) in hPGCs in vitro (*Figure 4f*, and *Figure 4—figure supplement 1e,h*, *Figure 4—figure supplement 1—source data 1*, *Figure 4—figure supplement 1—source data 2*) to decrease or increase nitrate influx in cells. We found that *Slc17a5* knockdown led to significantly reduced cell proliferation, and *Slc17a5* overexpression resulted in a small, but significant, increase in proliferation (*Figure 4g and h*). These findings indicated that sialin-mediated nitrate transportation promoted hPGCs proliferation. Cell cycle analysis indicated that *Slc17a5* knockdown prolonged the G1 phase, and shortened the S and G2/M phases (*Figure 4i*), resulting in G1 to S phase arrest. These results suggested that sialin-mediated nitrate transportation promoted hPGCs proliferation through the regulation of cell cycle transitions. Furthermore, *Slc17a5* knockdown resulted in increased apoptosis (*Figure 4j*).

Cells derived from the submandibular glands of *Slc17a5* knockdown mice (sgRNA two-cell embryo) showed decreased nitrate secretion and exhibited a significantly lower proliferation rate that that of cells derived from wild-type (WT) mice (*Figure 4k and l*); this further supports the conclusion that sialin-mediated nitrate transportation promoted cell proliferation.

We further investigated whether nitrate or sialin played a dominant role in IR protection. SialinH183R mutation has a defect in nitrate transportation; we overexpressed sialinH183R (*Figure 4m*) and found that sialinH183R could not increase cell proliferation similar to WT sialin (*Figure 4n*). These results demonstrated that nitrate could increase sialin expression in parotid gland cells, creating a nitrate-sialin feedback loop, and that sialin-mediated nitrate transportation is essential to maintain cell survival, promote parotid gland cell proliferation, and inhibit apoptosis. Sialin acts as a transporter in IR protection in which nitrate plays a dominant role. It is the sialin-meditated nitrate transportation that protects salivary glands from IR damage.

## Sialin-mediated nitrate transportation is closely related to IR damage in vitro

IR of hPGCs significantly impaired proliferation (*Figure 5a–c*), and increased apoptosis (*Figure 5d*). Sialin expression progressively and significantly decreased in hPGCs over time following IR, which showed that sialin is markedly affected by IR (*Figure 5e* and *Figure 5—source data 1*).

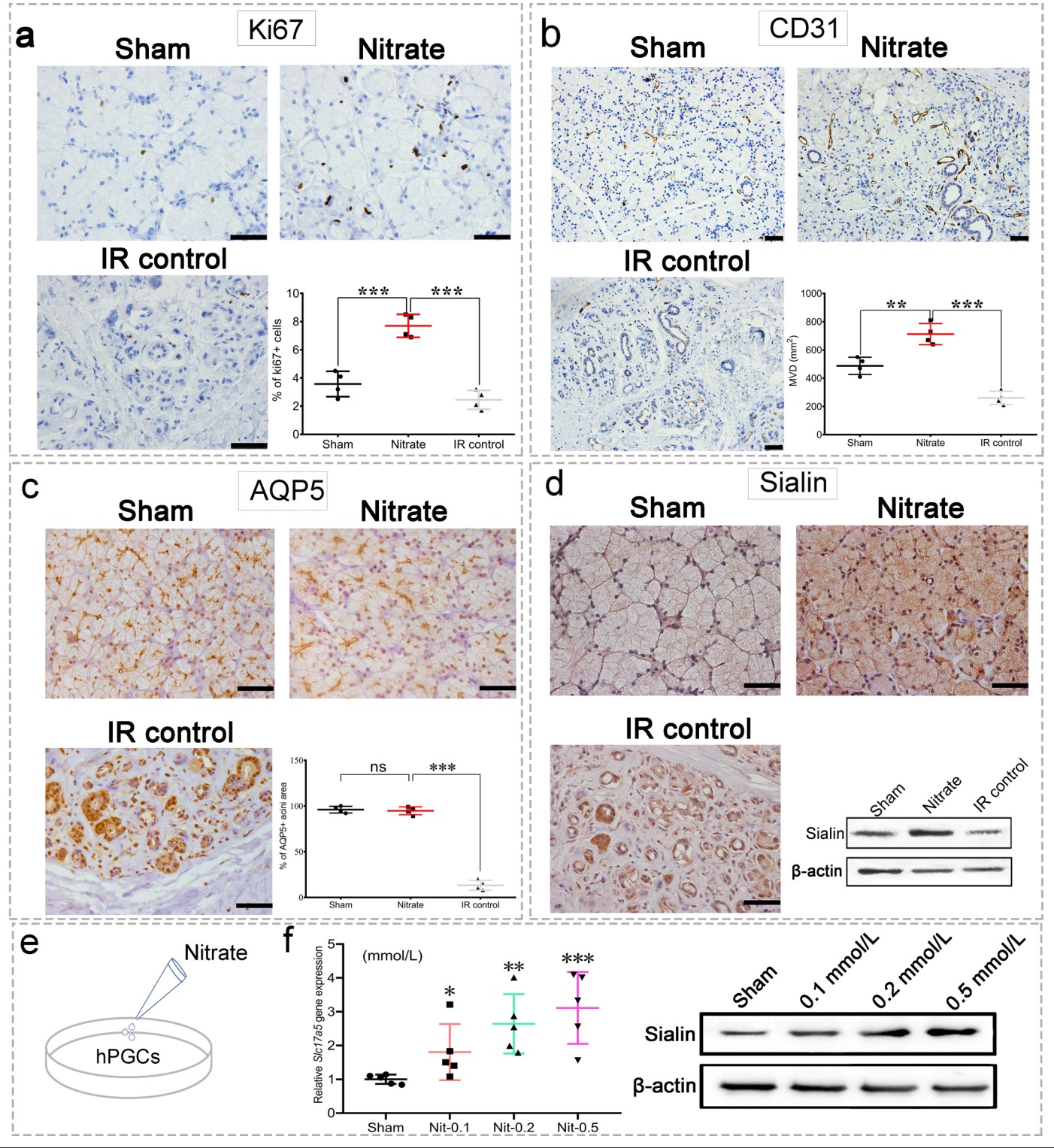

**Figure 3.** Changes in the parotid gland following irradiation (IR). (**a–d**) Immunohistochemical staining of parotid gland sections from miniature pigs in the sham, nitrate (2 mmol/kg·day), and IR control groups (n=4 animals per group; see *Figure 2a* for in vivo experimental design). Scale bar: 50 µm. Because the results for all nitrate doses showed similar trends (data not shown), we used data from the highest dose (2 mmol/kg·day) as representative of all doses. Nitrate group versus the sham group or the IR control group (**a–c**). (**a**) Cell proliferation rate, measured by Ki67 expression. (**b**) Microvessel density, measured by CD31 expression. (**c**) Aquaporin 5 (AQP5) expression; the areas of AQP5-positive cells among the groups are compared in the graph. (**d**) Immunohistochemical staining and western blot analysis of sialin expression.(**e**) Diagram of nitrate administration to human parotid gland cells

*Figure 3 continued on next page*

*Figure 3 continued*

(hPGCs) (in vitro, under physiological circumstances). (**f**) Sialin expression in hPGCs at 72 h after addition of different nitrate doses to culture medium (0.1, 0.2, or 0.5 mmol/L), measured by RT-PCR (left) and western blot (right); sham: blank control. n=5 per group. Data are expressed as the mean ± SEM. *p<0.05, **p<0.01, ***p<0.005, ns, no significance. RT-PCR, reverse transcription PCR; SEM, standard error of the mean.

The online version of this article includes the following source data for figure 3:

**Source data 1.** Western blot data for *Figure 3d*.

**Source data 2.** Western blot data for *Figure 3e*.

*Slc17a5* knockdown with siRNA and subsequent 5 Gy IR (*Figure 5—figure supplement 1a, b*, *Figure 5—figure supplement 1—source data 1*) significantly impaired cell proliferation (*Figure 5—figure supplement 1c,d*) and increased apoptosis (*Figure 5—figure supplement 1e*). Conversely, *Slc17a5* overexpression (*Figure 5e, f, g*, *Figure 5—source data 2*) that facilitated nitrate transportation preserved cell proliferation (*Figure 5h and i*) and counteracted IR-induced cell apoptosis (*Figure 5j*).

These results suggested that reduced sialin expression worsened IR damage, and that maintenance of sialin expression for nitrate transportation could promote cell proliferation and inhibit IR-induced apoptosis.

## Nitrate-mediated cytoprotection via the EGFR–AKT–MAPK signaling pathway

After nitrate administration, *EGFR* was significantly upregulated, which indicated that the EGFR signaling pathway was activated (*Figure 6a* and *Figure 6—source data 1*). Protein kinase B (AKT) and mitogen-activated protein kinase (MAPK) are downstream effectors of the EGFR pathway. They participate in maintenance of cell survival and promotion of cell proliferation, respectively. Cell proliferation is triggered by the MAPK pathway through activation of extracellular regulated protein kinase (ERK). We found that nitrate administration increased phosphorylation of EGFR, AKT, and ERK (*Figure 6a* and *Figure 6—source data 1*). Blockade of EGFR attenuated nitrate-mediated promotion of cell proliferation (*Figure 6b*). Moreover, blockade of EGFR inhibited phosphorylation of AKT and ERK; nitrate administration reversed this effect (*Figure 6c* and *Figure 6—source data 2*), which indicated that EGFR was upstream of both the AKT and MAPK signaling pathways. Notably, knockdown of sialin prevented nitrate-induced activation of EGFR suggesting that nitrate-mediated regulation of the EGFR signaling pathway required sialin (*Figure 6a* and *Figure 6—source data 3*).

Nitrate could be metabolized into nitrite ($NO_2^-$) and nitric oxide (NO). PTIO (2-Phenyl-4,4,5,5-tetramethylimidazoline-3-oxide-1-oxyl) is the NO scavenger. Sialin expression was decreased after NO was blocked by PTIO (*Figure 6d* and *Figure 6—source data 4*). We also found that EGFR phosphorylation was decreased after NO scavenging (*Figure 6d* and *Figure 6—source data 4*). We have demonstrated that nitrate increases sialin expression and EGFR phosphorylation via the NO pathway and thus regulates EGFR-AKT-MAPK signaling pathway.

Moreover, we have also observed significantly elevated phosphorylation of EGFR, AKT, and ERK in the nitrate group in vivo compared with the IR control and sham groups that were consistent with our in vitro study (*Figure 6e* and *Figure 6—source data 5*).

These results suggested that a nitrate-sialin feedback loop could mediate cell proliferation to protect the parotid gland from IR damage via the EGFR–AKT–MAPK signaling pathway (*Figure 6f*).

## Discussion

In this study, we demonstrated that dietary nitrate administration protected the parotid gland against IR damage in a dose-dependent manner, by using an established miniature pig model of IR-induced parotid gland hypofunction. We found that nitrate supplementation enhanced hPGCs cell proliferation via EGFR–AKT–MAPK signaling pathway. Therefore, dietary nitrate administration may be an effective novel approach for the prevention of IR-induced xerostomia.

In the present study, nitrate increased serous acinar and ductal cell proliferation rates, and increased MVD. Each of these parameters is crucial for maintenance of salivary gland homeostasis and restoration of function (*Aure et al., 2015*; *Ekström et al., 2017*; *Weng et al., 2018*). We showed a close

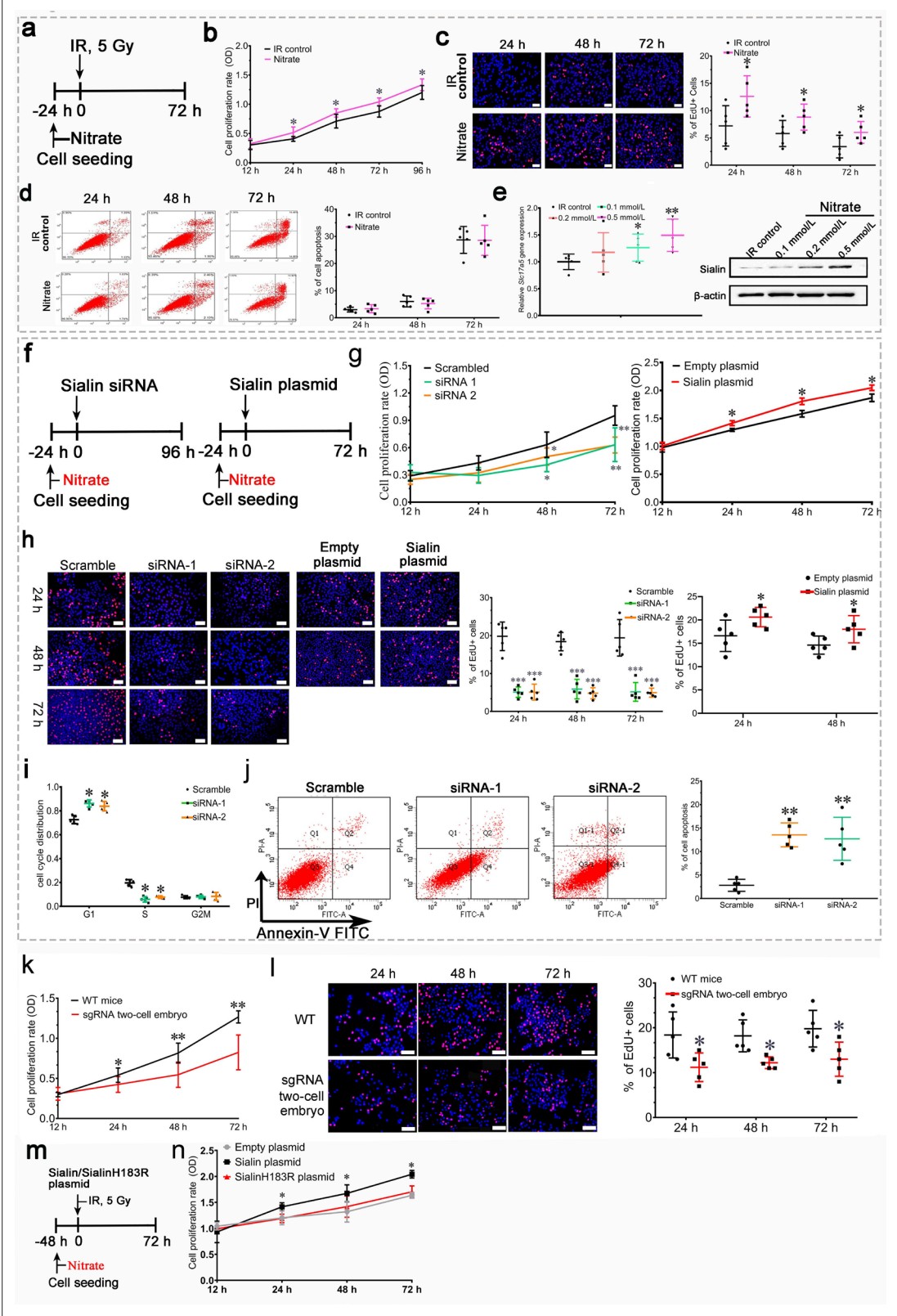

**Figure 4.** Effects of nitrate and sialin on salivary gland cell proliferation and apoptosis. (**a–e**) Effects of nitrate on human parotid gland cells (hPGCs). (**a**) Schematic of experimental design for nitrate administration to hPGCs. (**b**, **c**) Cell proliferation rate, determined using CCK-8 assay (**b**) or EdU staining (**c**). (**d**) Cell apoptotic rate, determined using flow cytometry. (**e**) Sialin expression in the IR control group and the nitrate groups at 72 h post-IR, determined using RT-PCR and western blot (representative time point; all remaining time points exhibited similar trends). (**a–d**) Addition of different

*Figure 4 continued on next page*

*Figure 4 continued*

nitrate doses (0.1, 0.2, or 0.5 mmol/L) to hPGCs culture medium prior to IR. 500 µM nitrate was used as the representative dose. All remaining doses exhibited similar trends. (**a–e**) Data are expressed as the mean ± standard error of the mean (SEM), n=5 culture plate replicates per group. *p<0.05, **p<0.01 for each nitrate group versus the IR control group. (**f–i**) Effects of sialin on hPGCs. (**f**) Schematic of experimental design for sialin siRNA or sialin plasmid delivery to hPGCs. (**g**) Cell proliferation rate determined using CCK-8 assay at 24, 48, and 72 h after siRNA or plasmid transfection. The 12 h time point was used as baseline, and scrambled siRNA and empty plasmid served as the control groups. (**h**) Cell proliferation rate determined by number of EdU+/Hoechst + cells at 24, 48, and 72 h after siRNA or plasmid transfection. Because plasmid transfection required 70% cell confluence, cell proliferation was determined at 24 and 48 h after plasmid transfection. (**i**) Cell cycle distribution analysis at 72 h (representative time point) after siRNA transfection. (**j**) Cell apoptotic rate, determined by flow cytometry, at 72 h after siRNA transfection (representative time point). *Slc17a5* overexpression did not affect apoptosis when hPGCs were cultured under normal conditions (data not shown). (**f–j**) Data are expressed as the mean ± SEM, n=5 culture plate replicates per group. *p<0.05, **p<0.01. (**k**, **l**) Submandibular gland cell proliferation compared between sgRNA two-cell embryo and wild-type (WT) mice. The expression level of sialin in sgRNA two-cell embryos was about 60 % of that in WT mice. (**k**) Cell proliferation rate determined using CCK-8 assay. (**l**) Cell proliferation rate, determined by number of EdU+/Hoechst + cells. (**k**, **l**) Data are expressed as the mean ± SEM, n=5 animals per group; *p<0.05, **p<0.01 for the sgRNA two-cell embryo group versus the WT group. (**m**, **n**) Nitrate or sialin has an effect on cell proliferation. (**k**) The experimental design scheme (sialinH183R is defective in nitrate transportation). (**l**) Cell proliferation rate, determined using CCK-8. Data are expressed as the mean ± SEM, n=5 culture plate replicates per group. *p<0.05, **p<0.01 for the sialin plasmid group versus the empty plasmid group. RT-PCR, reverse transcription PCR.

The online version of this article includes the following source data and figure supplement(s) for figure 4:

**Source data 1.** Western blot data for *Figure 4e*.

**Figure supplement 1.** Effects of nitrate on human parotid gland cells (hPGCs) under physiological conditions and transfection efficiencies of sialin siRNA and plasmids to hPGCs.

**Figure supplement 1—source data 1.** Western blot for *Figure 4—figure supplement 1f*.

**Figure supplement 1—source data 2.** Western blot for *Figure 4—figure supplement 1h*.

interaction between nitrate and sialin in vivo using a miniature pig model and in vitro using hPGCs, demonstrating a nitrate-sialin feedback loop in which the nitrate increased sialin expression and sialin facilitated nitrate influx into cells. It is the sialin-mediated nitrate transportation that plays a critical role in maintaining proliferation and survival of parotid gland epithelial cells; nitrate plays a dominant role in preventing IR damage and sialin acts as a nitrate transporter. Preventive nitrate administration increased sialin expression, promoted acinar and ductal cell proliferation, and reduced apoptosis via the EGFR–AKT–MAPK signaling pathway. However, nitrate administration could not reverse IR-induced damage when sialin expression was sharply reduced. These findings explained why preventive, but not therapeutic, nitrate administration was effective in our in vivo large animal study.

Several treatments have shown beneficial effects against IR-induced xerostomia, such as antioxidants (e.g., amifostine), gene transfer of human aquaporin 1 cDNA (*hAQP1*) (*Baum et al., 2017*; *Baum et al., 2009*; *Delporte et al., 1997*; *Gao et al., 2011*; *Shan et al., 2005*; *Vitolo and Baum, 2002*), fibroblast growth factor-2 (*Guo et al., 2014*), or sonic hedgehog (*Hu et al., 2018*), intraperitoneal injection of the immune inhibitor rapamycin (*Zhu et al., 2016*), stem cell transplantation, and salivary gland regeneration. Clinical trials showed positive effects of *hAQP1* gene transfer therapy (*Alevizos et al., 2017*; *Baum et al., 2012*; *Baum et al., 2010*), which exhibited beneficial effects in patients with established xerostomia (i.e., when the salivary glands already were severely damaged), but did not totally resolve IR induced xerostomia. However, cost, uncertain efficacy, potential side effects, tolerance issues, immunogenicity, and/or ethical issues have delayed translation of these therapeutic strategies to the clinic (*Grundmann et al., 2009*; *Vissink et al., 2015*). Our study showed that administration of inorganic nitrate significantly blunted IR-induced parotid gland hypofunction when administered prior to IR. In miniature pigs, nitrate administration resulted in preservation of approximately 85 % of secretory function and maintenance of nearly normal morphology at the highest dose provided in this study (2 mmol/kg·day). In addition, nitrate administration contributed to maintenance of levels of saliva constituents (e.g., amylase). Furthermore, inorganic nitrate lacks immunogenicity, is not subject to tolerance, is not toxic, and does not induce significant side effects at the doses employed (*Omar et al., 2012*).

Although nitrate is a natural substance present in the human diet, it has been regarded as a potential carcinogen for decades, and is generally believed to be harmful. However, recent studies failed to provide evidence for this assumption (*Bryan et al., 2012*; *Hezel et al., 2015*; *Khambata et al., 2017*; *McNally et al., 2016*; *Wu et al., 2013*; *Xie et al., 2016*). In 2011, the World Health Organization

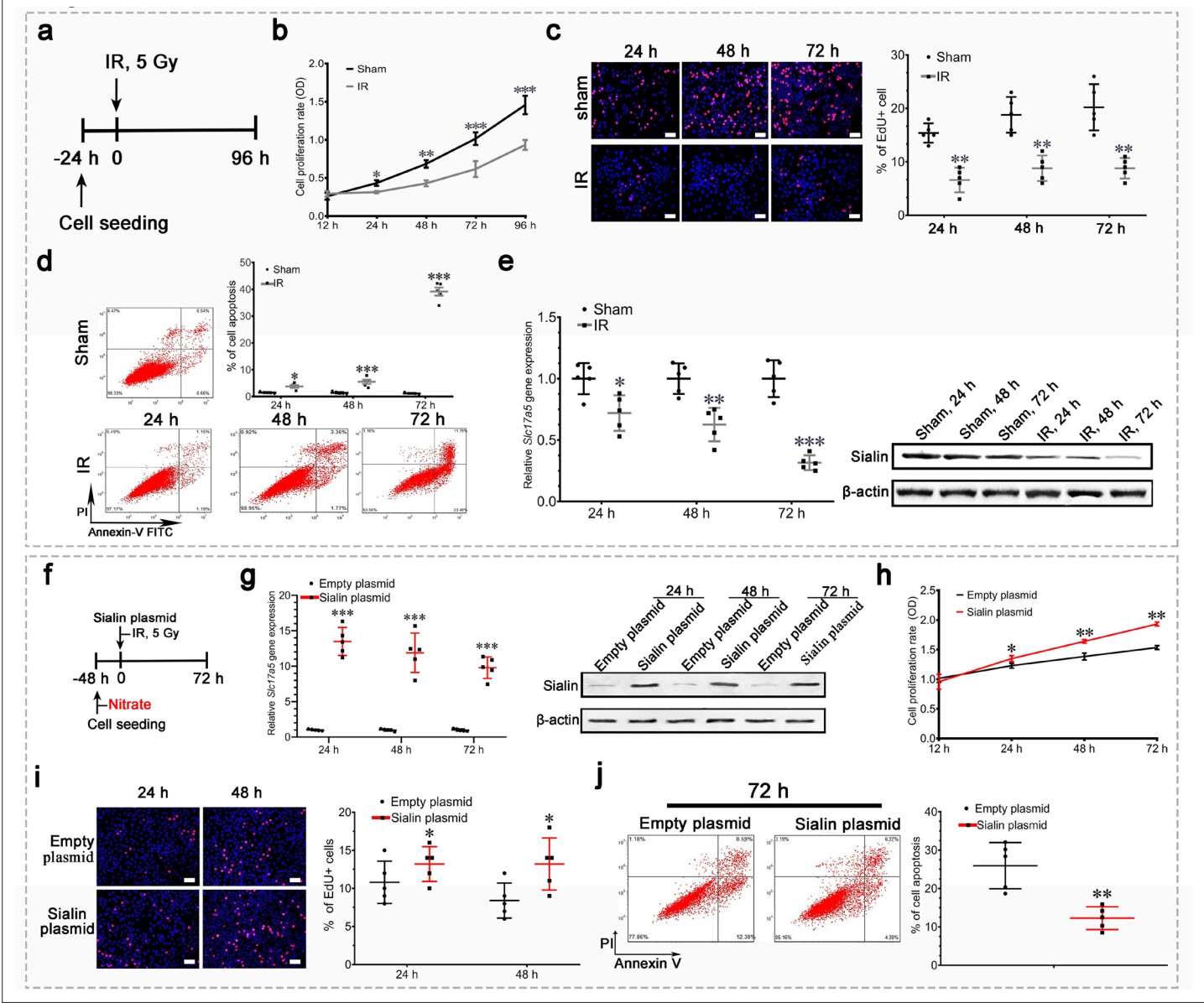

**Figure 5.** Sialin overexpression promoted cell proliferation and reduced cell apoptosis after irradiation (IR) in vitro. (**a–e**) Proliferation, apoptosis, and sialin expression of hPGCs cultured under physiological and IR conditions. (**a**) Schematic of experimental design. (**b**, **c**) Cell proliferation rate, determined using CCK-8 (**b**) or EdU assay (**c**). (**d**) Cell apoptotic rate, determined using flow cytometry. (**e**) Sialin expression, measured using reverse transcription PCR (RT-PCR; left) and western blot (right). (**b–e**) Sham group: hPGCs received 0 Gy; IR group: hPGCs received 5 Gy, n=5 culture plate replicates per group. Data are expressed as the mean ± SEM. *p<0.05, **p<0.01, ***p<0.005. (**f–j**) Effects of sialin overexpression on cell proliferation and apoptosis. (**f**) Schematic of experimental design. (**g**) Sialin expression in the sialin plasmid and empty plasmid groups, determined using RT-PCR and western blot. (**h**), (**i**) Cell proliferation rate, detected using CCK-8 assay (**h**) or EdU staining (**i**). (**j**) Cell apoptotic rate, determined using flow cytometry assay, at 72 h post-IR. (**g–j**) Data are expressed as the mean ± SEM, n=5 culture plate replicates per group. *p<0.05, **p<0.01, ***p<0.005 for the sialin plasmid group versus the empty plasmid group. SEM, standard error of the mean.

The online version of this article includes the following source data and figure supplement(s) for figure 5:

**Source data 1.** Western blot for *Figure 5e*.

**Source data 2.** Western blot for *Figure 5g*.

**Figure supplement 1.** Reduction of sialin expression impaired cell proliferation and increased apoptosis after irradiation (IR).

**Figure supplement 1—source data 1.** Western blot for *Figure 5—figure supplement 1b*.

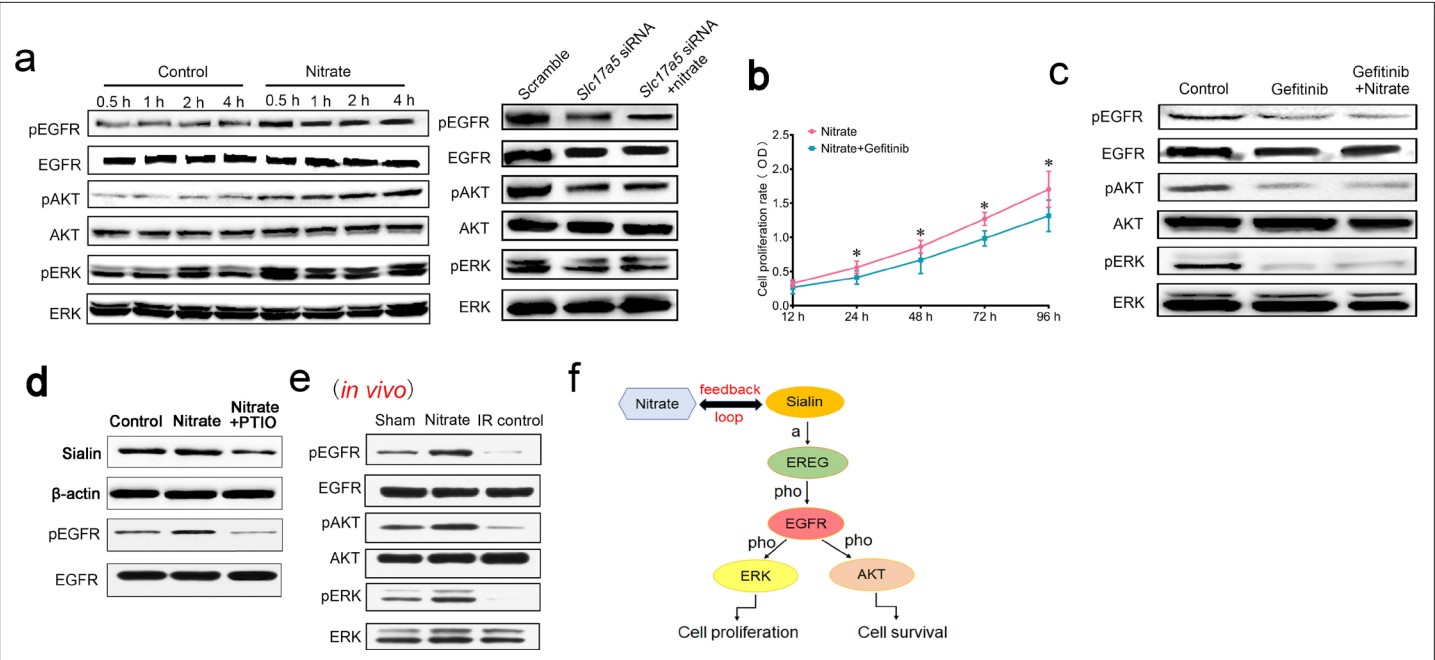

**Figure 6.** Mechanism of nitrate regulation of cell proliferation. (**a**) Phosphorylation of EGFR, AKT, and ERK following nitrate administration for 0.5, 1, 2, and 4 h, and phosphorylation of EGFR, AKT, and ERK following sialin knockdown for 24 h, n=5 sample replicates per group. (**b**) Cell proliferation rate after nitrate administration plus blockade of EGFR, determined using CCK-8 assay at 24, 48, 72, and 96 h. The 12 h time point was used as baseline. Data are expressed as the mean ± SEM, n=5 sample replicates per group. *p<0.05 for the nitrate plus gefitinib group versus the nitrate group. (**c**) Phosphorylation of EGFR, AKT, and ERK following EGFR blockade or nitrate administration plus EGFR blockade for 24 h, n=5 sample replicates per group. (**d**) Sialin and phosphorylation of EGFR expression after blocking nitrate-nitrite-nitric oxide (NO) pathway by PTIO. (**a–d**) Nitrate administration dose: 0.5 mmol/L. (**e**) Phosphorylation of EGFR, AKT, and ERK in parotid gland tissues of sham, nitrate, and IR control groups, in vivo study. (**f**) Schematic of the proposed nitrate–sialin feedback loop in radioprotection of salivary glands (a: activation; pho: phosphorylation). SEM, standard error of the mean.

The online version of this article includes the following source data for figure 6:

**Source data 1.** Western blot data for *Figure 6a* (the left part).

**Source data 2.** Western blot data for *Figure 6c*.

**Source data 3.** Western blot for *Figure 6a* (the right part).

**Source data 4.** Western blot data for *Figure 6d*.

**Source data 5.** Western blot data for *Figure 6e*.

published a guideline regarding nitrate and nitrite in drinking water which stated that nitrate is not carcinogenic, based on both laboratory animal studies and epidemiological studies (*World Health Organization, 2011*).

Nitrate occurs in two forms: organic and inorganic. Dietary nitrate is in the inorganic form (*Xia et al., 2015*). Although organic nitrate has been used for the treatment of cardiovascular disease (*Omar et al., 2016*), it has several notable disadvantages that limit its application in the clinic (*Omar et al., 2012*), which include poor absorption, low oral bioavailability due to first-pass metabolism in the liver, better effectiveness via sublingual, transdermal, or rectal administration, short half-life, development of tolerance resulting from long-term administration, and side effects such as headache and postural hypotension. Conversely, inorganic nitrate has a simple ionic structure that is easily absorbed, exhibits high oral bioavailability, has a long half-life, and causes no side effects (*Khambata et al., 2017*; *Qin et al., 2012*; *Xia et al., 2015*). Therefore, inorganic nitrate appears to comprise a safe and effective treatment for IR-induced salivary gland hypofunction.

The best dose of sodium nitrate used in the miniature pig in the in vivo study was 2 mmol/kg·day. The coefficient of drug administration from pigs to humans was 0.73; thus, the dose converted to human was 1.46 mmol/kg·day. For example, an adult weighing 60 kg might have to take 7 g sodium nitrate per day, which is considered a relatively high dose. However, the parotid gland of miniature pigs received 60 Gy IR doses in our study, and the period of irradiation lasted for only 5 days receiving

7.5 Gy each day. When HNC patients receive IR, the IR period lasts for two months and they receive only 2 Gy daily, and the total dose would not be given to the parotid gland as most doses are focused on the tumor. In this case, the parotid gland of patients with HNC could not receive as strong a dose as the experimental animals in this study. Taken together, although the nitrate intake from diet food was 3–7 mg/kg per day for clinical treatment (an adult weighing 60 kg takes 0.18–0.42 g sodium nitrate per day) (*Ashworth and Bescos, 2017*), it is considered that the dose for clinical treatment could be well tolerated by humans.

Dietary nitrate can be metabolized into nitric oxide (NO) via the nitrate–nitrite–NO pathway (*Lundberg et al., 2008*). NO is an important gaseous signaling molecule that participates in many physiological processes, including stimulation of reparative angiogenesis and reduction of oxidative stress (*Sengupta et al., 2004*; *Sessa, 2009*). Under hypoxic or acidic conditions, NO synthesis primarily occurs via the nitrate–nitrite–NO exogenous pathway. Notably, IR induces a hypoxic and acidic environment within salivary glands. Dietary nitrate supplementation results in NO production and reduces hypoxia by inducing a prolonged increase in blood flow (*Lundberg et al., 2008*), and increased gland microvascularization, which is beneficial for production of saliva by acinar cells. Furthermore, nitrate-mediated NO formation also increases sialin expression and upregulates EGFR-AKT-MAPK signaling pathways, the classical pathways that are responsible for promoting cell proliferation, maintaining cell survival, and preventing cell apoptosis (*Sabbah et al., 2020*; *Sun et al., 2015*; *Zhang et al., 2011*).

However, nitrate is a stable substance that cannot be all metabolized to NO under IR, while sialin expression increases after nitrate administration, which helps sialin transport more nitrate into cells. It is considered that, except for NO, there are other mechanisms involved in preventing IR damage. It has been revealed that nitrate is related to mitochondrial respiration, activation of key metabolic regulatory pathways, and reduction of oxidative stress (*Lundberg et al., 2011*; *Lundberg et al., 2018*). Thus, more detailed mechanisms, including nitrate regulation of mitochondrial functions, should be the focus of future studies.

In conclusion, nitrate administration preserves miniature pig parotid gland function and protects the parotid gland against IR damage. The underlying mechanism involves the sialin-mediated nitrate transportation that maintains salivary gland homeostasis and prevents IR damage via the EGFR–AKT–MAPK signaling pathways. Dietary nitrate is therefore a potential treatment modality for IR-induced salivary gland hypofunction and it appears worthwhile to test in clinical studies.

## Materials and methods

**Key resources table**

| Reagent type (species) or resource | Designation | Source or reference | Identifiers | Additional information |
|---|---|---|---|---|
| Gene (*Sus scrofa*) | Sialin | NCBI | NC-010443.5 | |
| Gene (*Homo sapiens*) | Sialin | NCBI | NC_000006.12 | |
| Strain, strain background (*S. scrofa*) | Wild-type | Chinese Agricultural University | RRID:Addgene_66988 | 8–12 months, male |
| Transfected construct (*H. sapiens*) | siRNA to sialin | Thermo Fisher Scientific | | 5'-TCCTGGAGGATATGTTGCCAGCAAA-3' 5'-CATCACAAATACATTTGCCACTATT-3' |
| Antibody | (Rabbit polyclonal) anti-ki67 | Abcam | ab15580 RRID:AB_443209 | IHC (1:200) |
| Antibody | (Rabbit polyclonal) anti-CD31 | Abcam | ab28364 RRID:AB_726362 | IHC (1:50) |
| Antibody | (Rabbit polyclonal) anti-AQP5 | Thermo Fisher Scientific | PA5-36529 RRID:AB_2553573 | IHC (1:50) |
| Antibody | (Rabbit polyclonal) anti-sialin | Thermo Fisher Scientific | PA5-42456 RRID:AB_2577049 | IHC (1:200) WB (1:500) |
| Antibody | (Rabbit monoclonal) anti-pEGFR | Cell Signaling Technology | #3777 S RRID:AB_2096270 | WB (1:1000) |

*Continued on next page*

*Continued*

| Reagent type (species) or resource | Designation | Source or reference | Identifiers | Additional information |
|---|---|---|---|---|
| Antibody | (Rabbit monoclonal) anti-EGFR | Cell Signaling Technology | #4267 S RRID:AB_2246311 | WB (1:1000) |
| Antibody | (Rabbit monoclonal) anti-pAKT | Cell Signaling Technology | #4060 S RRID:AB_2315049 | WB (1:1000) |
| Antibody | (Rabbit monoclonal) anti-AKT | Cell Signaling Technology | #9272 S RRID:AB_2246311 | WB (1:1000) |
| Antibody | (Rabbit monoclonal) anti-pERK | Cell Signaling Technology | #4370 S RRID:AB_2315112 | WB (1:1000) |
| Antibody | (Rabbit monoclonal) anti-ERK | Cell Signaling Technology | #4695 S RRID:AB_390779 | WB (1:1000) |
| Recombinant DNA reagent | Sialin plasmid (pHS-LW066 vector) | Obio Technology | | Wild-type |
| Recombinant DNA reagent | SialinH183R plasmid (pHS-LW066 vector) | Obio Technology | | Mutant type does not transport nitrate |
| Commercial assay or kit | Total Nitric Oxide and Nitrate/Nitrite Parameter Assay Kit | R&D Systems | PKGE001 | |
| Software, algorithm | Prism 6 | GraphPad | https://www.graphpad.com/; RRID:SCR_002798 | |

## Animals

Healthy miniature pigs (8–12 months of age, 40–60 kg body weight) were purchased from the Institute of Animal Science of Chinese Agricultural University (Beijing, China). All animals were maintained under conventional conditions with free access to water and food. Food stock (200–250 g, mixed with water) was supplied twice per day, at 08:30 and 17:00. Animal studies were conducted according to the NIH's Guide for the Care and Use of Laboratory Animals, and approved by the Animal Care and Use Committee of Capital Medical University (Beijing, China; approval no. AEEI-2015-098).

## Single-dose IR of the parotid gland

### Single-dose IR

One parotid gland of each miniature pig received a single dose of 20 Gy IR (biological dose equal to 60 Gy) using a three-dimensional (3D) conformal technique (*Figure 2—figure supplement 1*), in accordance with our previously established model (*Guo et al., 2014*; *Li et al., 2005*; *Shan et al., 2005*). Briefly, animals were first anesthetized with a combination of ketamine chloride (6 mg/kg) and xylazine (0.6 mg/kg) injected intramuscularly. A radiation field of one parotid gland and 20 Gy was directed to the targeted parotid gland using a linear accelerator (SL 7520; Philips Medical Systems Inc, Bothell, WA) with 6 mV of photon energy at 3.2 Gy/min. The contralateral gland was outside of the radiation field and received less than 1 Gy.

### Groups

Sixteen miniature pigs were randomly divided into four groups (n=4 each): one naive control ('sham') group, one IR control group, and two nitrate experimental groups (preventive and therapeutic). The sham group received 0 Gy, while the IR control group and nitrate groups received 20 Gy each.

### Nitrate administration

Sodium nitrate (1 mmol/kg, Beijing Yili Fine Chemicals, Beijing, China) was added to the food twice per day (2 mmol/kg·day). In the prevention group, nitrate was administered beginning 1 -week pre-IR, and then consecutively for 4 months (*Figure 4a*). In the therapeutic group, nitrate administration was started 2 months post-IR, and then consecutively for 4 months (*Figure 4d*).

## Fractionated IR of the parotid gland

Fractionated IR

Fractionated IR, which better mimics the clinical situation than single-dose IR, was also used to study the effects of nitrate administration on IR damage to the parotid gland. One parotid gland of each miniature pig received fractionated IR—7.5 Gy for 5 consecutive days (biological dose equal to 60 Gy)—in accordance with our previously established model (*Gao et al., 2011*; *Hu et al., 2018*; *Zhu et al., 2016*). All procedures were conducted with pigs under general anesthesia, as described above. Briefly, axial computerized tomographic scans were employed to determine the IR plan using a 3D treatment planning system (Pinnacle3, version 7.6; ADACInc, Concord, CA). Calculations showed that more than 95 % of the IR dose covered the entire target volume of the parotid gland. The reference point for all dose calculations was the center-targeted parotid gland. The technology of image-guided radiation therapy (IGRT) was employed during IR. Real-time images of the parotid gland area were obtained with iViewGT (Elekta AB, Stockholm, Sweden) and was checked to ensure consistency with the predetermined IR plans before the imaging workflow was integrated into the IGRT system prior to IR. Next, animals were irradiated with 6 mV of photon energy at 3.2 Gy/min with the Elekta Synergy accelerator (Elekta AB). The contralateral gland was outside of the radiation field and received less than 1 Gy.

Groups

Twenty-four pigs were randomly divided into six groups (n=4): one naive control ('sham') group, one IR control group, and four nitrate experimental groups (different doses). The sham group received 0 Gy, while the IR control group and nitrate groups received 7.5 Gy for 5 consecutive days each.

Nitrate administration

Different doses of sodium nitrate (1, 0.5, 0.25, or 0.125 mmol/kg) were added to the food of the four treatment groups, twice per day (2, 1, 0.5, and 0.25 mmol/kg·day, respectively). Nitrate was administered beginning 1 -week pre-IR, and then consecutively for 4 months (*Figure 5a*).

## Parotid gland saliva and blood sample collection

Parotid gland saliva and blood samples were collected at different time points (*Figure 1A and D*; *Figure 2A*). Briefly, miniature pigs first received general anesthesia, after which pilocarpine (0.1 ml/ kg body weight) was injected intramuscularly to stimulate parotid gland saliva secretion. A modified Lashley cup was used to collect saliva for 10 min. Saliva flow rates were expressed as volume (ml) per 10 min per gland. Blood samples were simultaneously collected from the precaval vein. The collected saliva and blood samples were analyzed by standard clinical chemistry and—for blood—hematology procedures, as previously described (*Gao et al., 2011*; *Zhu et al., 2016*).

## Local blood flow measurement

Blood flow rate of the targeted parotid gland was measured after saliva collection. In the parotid gland area, three, aseptic 2.5 mm deep holes were made using an epidural anesthesia needle. Thereafter, local blood flow rate was measured aseptically using a 3 -mm laser Doppler blood flow probe (Moor Instruments Ltd, Axminster, UK) for 3 min, at each of the three positions (*Guo et al., 2014*; *Xu et al., 2010*).

## Measurement of salivary and serum nitrate concentrations

Saliva and blood samples were immediately centrifuged at 500 ×*g* for 20 min at room temperature. Nitrate concentrations were detected using the Total Nitric Oxide and Nitrate/Nitrite Parameter Assay Kit (PKGE001; R&D Systems, Minneapolis, MN), following standard experimental procedures provided by the manufacturer.

## Histological and immunohistochemical assays

Animals were sacrificed at the end points of our experimental designs (*Figure 4a and d*; *Figure 5a*). Parotid glands were collected, fixed in 4 % paraformaldehyde, dehydrated in gradient ethanol solutions, embedded in paraffin, and sectioned at 4 μm thickness. H&E staining was performed to analyze

morphological changes. Immunohistochemical staining was employed to determine the expression of CD31 (Abcam, Cambridge, UK, RRID:AB_726362), Ki67 (Abcam, RRID:AB_443209), AQP5 (Thermo Fisher Scientific, Waltham, MA, RRID:AB_2553573), and sialin (Thermo Fisher Scientific, RRID:AB_2577049). CD31 and Ki67 were used to analyze the MVD and cell proliferation, respectively. AQP5 is expressed in apical membranes of acinar cells and plays a key role in saliva secretion. Thus, AQP5 expression was assessed to determine the saliva secretory function of the parotid gland on a histological level.

## TUNEL in situ assay

To analyze cell apoptosis in parotid gland tissue, a terminal deoxynucleotidyl transferase dUTP nick-end labeling (TUNEL) in situ assay was performed, in accordance with the manufacturer's instructions (RiboBio, Guangzhou, China). In brief, parotid gland sections were incubated with proteinase K (20 μg/ml) at 37 °C for 30 min, and subsequently subjected to terminal deoxynucleotidyl transferase (TdT) and Alexa Fluor 567-conjugated EdUTP at 37 °C for 1 hr. Nuclei were stained with Hoechst 33,342 (RiboBio) for 30 min. Stained sections were assessed by confocal microscopy (Olympus, Tokyo, Japan).

## Cell cultures and IR

### Human parotid gland cells

A normal parotid gland biopsy specimen was collected from a 38-year-old female patient, who underwent superficial parotidectomy for a benign tumor, and was used for culturing primary human parotid gland epithelial cells. Briefly, the specimen was harvested during surgery, carefully avoiding the tumor. The tissue was finely chopped, and then digested by collagenase I (1 mg/ml; Sigma-Aldrich, Shanghai, China) and dispase (1 mg/ml; Sigma-Aldrich) at 37 °C for 30 min. Digested cells were centrifuged at 168 ×g for 5 min, washed two times with phosphate-buffered saline (PBS), and then suspended in serum-free DMEM-F12 culturing medium (Thermo Fisher Scientific), and the medium was changed every 2 days. Passages 2–6 of hPGCs were used in this study. Human parotid gland biopsy sample collection and use for research purpose were approved by the ethics committee of Beijing Stomatological Hospital, Capital Medical University (Beijing, China; approval no. CMUSH-IRB-KJ-PJ-2019-06 F). The patient has provided written informed consent prior to sample collection.

To explore the contribution of nitrate to alleviation of IR damage in vitro, hPGCs were subjected to 5 Gy IR. Medium was changed post-IR, and cells were cultured for an additional 72–96 h.

### Nitrate administration to irradiated hPGCs

Different nitrate doses (0.1, 0.2, or 0.5 mmol/L, Sigma-Aldrich) were added to the culture medium of hPGCs after 5 Gy IR to determine if nitrate can abrogate IR damage to cells. These doses did not change the buffer system of the medium, and were not toxic to the cells (data not shown).

### Nitric oxide blocking

In order to block nitric oxide expression, PTIO (2-Phenyl-4,4,5,5-tetramethylimidazoline-3-oxide-1-oxyl, 5 mmol/l, Sigma-Aldrich) was added to cultures of hPGCs.

### EGFR signaling pathway blocking

In order to block the EGFR signaling pathway, gefitinib (0.2 μm/L, Selleck, Shanghai, China) was added to cultures of hPGCs.

## Mouse submandibular gland cells

C57 mice were generated in which the gene for sialin (*Slc17a5*) was knocked out by two-cell embryo microinjection (sgRNA two-cell embryo), as previously described (*Wang et al., 2017*; *Wu et al., 2019*). Submandibular glands were dissected from the sgRNA two-cell embryo and WT mice; the gland cells were cultured in vitro, following the same procedure as for hPGCs described above. Animal studies were conducted according to the NIH's Guide for the Care and Use of Laboratory Animals, and approved by the Animal Care and Use Committee of Capital Medical University (Beijing, China; approval no. AEEI-2017-009).

## Cell proliferation assays

Proliferation rates of hPGCs and cells derived from mouse submandibular glands were measured with a 5'-ethynyl-2'-deoxyuridine (EdU) Staining Kit (RiboBio) and with Cell Counting Kit-8 (CCK-8; Dojindo, Shanghai, China) respectively, in accordance with the manufacturers' instructions.

For the EdU assay, cells were incubated with EdU solution in a 24-well plate for 2 h, fixed in 4 % paraformaldehyde at room temperature for 30 min, rinsed with PBS, and stained with Apollo solution (RiboBio). Nuclei were stained with Hoechst 33,342. Hoechst-stained cells and EdU-positive cells were counted using Image-Pro Plus 6.0 software (Media Cybernetics, Bethesda, MD).

For the CCK-8 assay, cells were incubated with 10 µl CCK-8 solution in a 96-well plate for 1 hr. Absorbance was measured at 450 nm by using a microplate reader (BioTek Instruments, Winooski, VT); the absorbance level at 12 hr served as baseline.

## Slc17a5 overexpression and knockdown in vitro

For gain- and loss-of-function approaches, *Slc17a5* was overexpressed or knocked down in hPGCs. In transfection experiments, a plasmid containing sialin full-length cDNA or sialinH183R mutant DNA, which did not transport nitrate (Obio Technology, Shanghai, China) was used for overexpression, whereas sialin siRNA (Invitrogen, Carlsbad, CA) was used for knockdown experiments. Empty plasmid (Obio Technology) and Scrambled Stealth siRNA (Invitrogen) served as the respective controls. Sialin siRNA nucleotide sequences were: siRNA-1, TCCTGGAGGATATGTTGCCAGCAAA, and siRNA-2, CATCACAAATACATTTGCCACTATT; the siRNA transfection protocol was previously described (*Feng et al., 2013*). Transfected cells were irradiated and supplemented with nitrate. For the plasmid transfection experiments, Lipofectamine 3000 Reagent (Invitrogen) was used, in accordance with the manufacturer's instructions. Briefly, Lipofectamine 3000 was diluted in Opti-MEM I Reduced Serum Medium (Invitrogen) in one tube and incubated at room temperature for 5 min. Plasmid was added to Opti-MEM I Medium (Thermo Fisher Scientific) in another tube. This diluted plasmid was then added to the diluted Lipofectamine 3000 solution and incubated at room temperature for 20 min. The final transfection complex was dripped gently onto plates with 70 % confluent hPGCs, and the medium was changed 24 hr after transfection.

## Reverse transcription PCR

Total RNA was extracted by using TRIzol Reagent (Thermo Fisher Scientific), and the residual DNA was removed by RNase-Free DNase (Promega, Madison, WI). Total RNA samples (1 µg) were reverse transcribed to synthesize cDNA by AMV Reverse Transcriptase (Promega), in accordance with the manufacturer's instructions. RT-PCR was performed by using an ABI Prism 7000 Sequence Detection System (Applied Biosystems, Thermo Fisher Scientific) with SYBR Green Reagent (Roche, Basel, Switzerland). The primer sequences used are listed in Table S1 in *Supplementary file 1*. Relative gene expression levels were calculated using the comparative cycle threshold method ($2^{-\Delta\Delta CT}$) and normalized to the level of *β-actin*.

## Western blot

Total protein was extracted by radioimmunoprecipitation assay (RIPA) lysis buffer (Applygen Technologies, Beijing, China), and the protein concentration was measured with a Bicinchoninic Acid (BCA) Protein Assay Kit (Thermo Fisher Scientific). Total protein samples (20 µg) were separated by 10 % sodium dodecyl sulfate polyacrylamide gel electrophoresis (SDS-PAGE), and transferred to polyvinylidene difluoride (PVDF) membranes (Millipore, Beijing, China). Membranes were blocked by 5 % skim milk and incubated with the following primary antibodies: rabbit anti-sialin (Thermo Fisher Scientific, RRID:AB_2577049), rabbit anti-phospho-EGF Receptor (Tyr1068) (Cell Signaling Technology [CST], Shanghai, China, RRID:AB_2096270), rabbit anti-EGF Receptor (D38B1) (CST, RRID:AB_2246311), rabbit anti-phospho-AKT (Ser473) (CST, RRID:AB_2315049), rabbit anti-AKT (CST, RRID:AB_2246311), rabbit anti-phospho-ERK1/2 (Thr202/Tyr204) (CST, RRID:AB_2315112), rabbit anti-ERK1/2 (CST, RRID:AB_390779), and mouse anti-β-actin (Abcam, RRID:AB_2305186). Thereafter, membranes were incubated with goat anti-rabbit or anti-mouse secondary antibodies (both Abcam) and visualized by enhanced chemiluminescence (BD Biosciences, Franklin Lakes, NJ).

## Flow cytometric assay

To analyze cell cycle and apoptosis, flow cytometric analysis was performed. Briefly, for cell cycle phase distribution analysis, cells were harvested, fixed in 70 % ethanol at 4 °C overnight, then washed with PBS and incubated with RNase A (Sigma-Aldrich) at room temperature for 30 min. Cells were then stained with propidium iodide (Thermo Fisher Scientific) and analyzed by flow cytometry.

Cell apoptosis was detected by using the Annexin V-FITC Apoptosis Detection Kit (BD Biosciences). In brief, cells were trypsinized and suspended with bonding buffer. After addition of Annexin V-FITC solution and incubation for 15 min, 5 µl propidium iodide was added, and analyzed by flow cytometry as above.

## Statistical analysis

The number of animals or cell cultures used is indicated for each experiment. All cell cultures were performed in duplicate. Data are presented as the mean ± SEM. A one-way ANOVA was used to analyze study results, assuming equal variances. The levels of statistical significance were as follows: $*p<0.05$, $**p<0.01$, $***p<0.001$; $\Delta<0.05$, $\Delta\Delta<0.01$. Statistical analysis was performed using SPSS 18.0 software (SPSS Inc, Chicago, IL).

## Acknowledgements

The authors sincerely thank Prof. Li Yu from Tsinghua University, Dr. Indu S Ambudkar, Dr. Bruce Baum, and Dr. Xibao Liu (Molecular Physiology and Therapeutics Branch and Secretory Physiology Section, NIDCR, NIH) for their scientific input. This study was supported by grants from the Chinese Research Unit of Tooth Development and Regeneration, CAMSI Innovation Fund for Medical Sciences, No. 2019–12 M-5-031; the National Natural Science Foundation of China (82030031, 91649124 and 81600883 to SW); Beijing Municipal Science & Technology Commission No. Z181100001718208; Beijing Municipal Education Commission No. 119207020201; Beijing Hospitals Authority of Hospitals' Mission Plan, code:SML20151401; Beijing Municipality Government grants (Beijing Scholar Program-PXM2018_014226_000021; PXM2018_193312_000006_0028S643_FCG, PXM2019_014226_000011, PXM2020_014226_000005; Z181100001718208).

## Additional information

### Funding

| Funder | Grant reference number | Author |
|--------|------------------------|--------|
| National Natural Science Foundation of China | 82030031 | Songlin Wang |
| National Natural Science Foundation of China | 91649124 | Songlin Wang |
| National Natural Science Foundation of China | 81600883 | Songlin Wang |
| Chinese Research Unit of Tooth Development and Regeneration, CAMSI Innovation Fund for Medical Sciences | No. 2019-12M-5-031 | Songlin Wang |
| Beijing Municipal Science & Technology Commission | No. Z181100001718208 | Songlin Wang |
| Beijing Municipal Education Commission | No. 119207020201 | Songlin Wang |
| Beijing Hospitals Authority of Hospitals' Mission Plan | code:SML20151401 | Songlin Wang |

The funders had no role in study design, data collection and interpretation, or the decision to submit the work for publication.

## Author contributions
Xiaoyu Feng, Data curation, Formal analysis, Investigation, Methodology, Project administration, Resources, Software, Validation, Visualization, Writing - original draft, Writing – review and editing; Zhifang Wu, Methodology, Project administration, Resources, Validation, Visualization, Writing - original draft; Junji Xu, Baoxing Pang, Xingmin Qu, Liang Hu, Luyuan Jin, Dengsheng Xia, Shimin Chang, Jingsong Wang, Methodology; Yipu Xu, Methodology, Project administration, Visualization; Bin Zhao, Methodology, Project administration; Lei Hu, Methodology, Software, Visualization; Zhipeng Fan, Conceptualization, Methodology; Chunmei Zhang, Funding acquisition; Songlin Wang, Conceptualization, Funding acquisition, Methodology, Supervision, Writing – review and editing

## Author ORCIDs
Songlin Wang (ID) https://orcid.org/0000-0002-7066-2654

## Ethics
Human subjects: Human parotid gland biopsy sample was obtained under a protocol approved by the ethics committee of Beijing Stomatological Hospital, Capital Medical University.
Animal studies were conducted according to the NIH's Guide for the Care and Use of Laboratory Animals, and approved by the Animal Care and Use Committee of Capital Medical University.

## Decision letter and Author response
Decision letter https://doi.org/10.7554/eLife.70710.sa1
Author response https://doi.org/10.7554/eLife.70710.sa2

---

# Additional files

## Supplementary files
Transparent reporting form
Supplementary file 1. Table 1 Reverse transcription- PCR primer sequences.

## Data availability
All data generated or analysed during this study are included in the manuscript and supporting files. Source data files have been provided for: Figure 1G, Figure 2E, Figure 3A–D, and Figure 6A, C, D and E, available on Dryad Digital Repository (https://doi.org/10.5061/dryad.fn2z34ttq).

The following dataset was generated:

| Author(s) | Year | Dataset title | Dataset URL | Database and Identifier |
|---|---|---|---|---|
| Feng X, Wu Z | 2021 | Dietary nitrate supplementation prevents radiotherapy-induced xerostomia | http://dx.doi.org/10.5061/dryad.fn2z34ttq | Dryad Digital Repository, 10.5061/dryad.fn2z34ttq |

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
