## [Decision Letter]

Thank you for submitting your article "Dietary nitrate supplementation prevents radiotherapy-induced xerostomia" for consideration by *eLife*. Your article has been reviewed by 2 peer reviewers, and the evaluation has been overseen by a Reviewing Editor and Wafik El-Deiry as the Senior Editor. The following individuals involved in review of your submission have agreed to reveal their identity: Fred Bunz (Reviewer #1); Changyu Zheng (Reviewer #2).

Essential revisions:

The reviewers were enthusiastic about the findings and their potential for practical application in the clinic to address xerostomia. Revisions are suggested by each of the reviewers, including more discussion of limitations of the evidence for the role of EGFR signaling, nitrate signaling and free radicals.

*Reviewer #1:*

Xerostomia, a frequent side-effect of radiotherapy to the head and neck, can compromise dental health, nutrition and speech. As a result, affected patients often experience a significant decline in their quality of life. Current measures used to prevent or manage radiation-induced xerostomia are often ineffective. In the present study, the authors set out to determine whether dietary supplementation with inorganic nitrate might be a useful approach to the prevention and/or management of this important manifestation of radiation toxicity.

The in vivo studies were based on the miniature pig, a previously established model of salivary gland dysfunction that recapitulates many of the radiation responses observed in human tissues. Dietary nitrate was administered either prophylactically, prior to irradiation, or therapeutically to irradiated animals. Pre-treatment with nitrate resulted in the preservation of salivary gland function, longitudinally assessed by salivary flow rate and local blood flow, in animals that were subsequently treated with clinically-relevant doses of ionizing radiation. Histologically, these pre-treated animals also exhibited elevated levels of acinar glad proliferation and lower levels of local fibrosis, a late effect of radiotherapy that is observed in human patients. In striking contrast, nitrate supplementation administered after irradiation did not appear to ameliorate these long-term toxicities.

Molecular analyses of parotid tissues supported these impressive clinical observations, as did in vitro studies of human parotid gland cells. An important finding was the dose-dependent relationship between nitrate administration and the expression of the transmembrane protein sialin. Sialin acts as a local transporter of nitrate and was shown here to enhance proliferation and suppress apoptosis in this cell type, in what the authors describe as a positive feedback loop. This novel hypothesis provides a plausible explanation for the observed clinical and histological effects of nitrate pre-treatment.

The mechanistic studies of how sialin might trigger cell proliferation were exclusively focused on the activity of the EGFR-AKT-MAPK pathway. The activation of this pathway by sialin and nitrate seemed real, but modest in magnitude. Compared to the other sections, this part was somewhat underdeveloped. From the data presented, it is difficult to conclude that EGFR activation would account for the dramatic clinical and histologic effects of pre-treatment. This is a relatively minor weakness, which could be addressed by adding further context to the discussion. Overall, this was a rigorous study that presents a new, highly translatable approach to radiation-induced xerostomia.

The question of whether the EGFR-AKT-MAPK is responsible for the observed effects could simply be addressed in a revised discussion. It would suffice to mention the caveat that additional pro-proliferative or anti-apoptotic mechanisms, not addressed in this study, could conceivably be involved.

Second, it might also be helpful to contextualize the doses of sodium nitrate used in vivo. Would similar dosage in humans be tolerable? What was the level of nitrates in the non-supplemented feed? In the average human diet?

*Reviewer #2:*

Manuscript of "Dietary nitrate supplementation prevents radiotherapy-induced xerostomia" by Xiaoyu Feng, et al., is an interesting article. The presented data suggest that nitrate, dietary nitrate supplementation, effectively prevented radiotherapy-induced-induced xerostomia via the EGFR-AKT-MAPK signaling pathway, suggesting that dietary nitrate supplementation may provide a novel, simple, safe and effective way to prevent IR-induce xerostomia. Their experimental design is straightforward. Most results are solid. This manuscript will be a useful reference for the field.

Following comments are needed to clarify.

1. Please check if "Mechanically" in line 26 is right word.

2. Would like to check if nitrate affects free radicals, which are produced during irradiation, and mitochondrial functions. If authors don't know yet, hope authors can try to understand these in their future study.

---

## [Author Response]

Reviewer #1:[…]The question of whether the EGFR-AKT-MAPK is responsible for the observed effects could simply be addressed in a revised discussion. It would suffice to mention the caveat that additional pro-proliferative or anti-apoptotic mechanisms, not addressed in this study, could conceivably be involved.

Thanks very much for the reviewer 1’s comments. We have now addressed that EGFR-AKT-MAPK is responsible for promoting cell proliferation and reducing cell apoptosis in the Discussion as you had suggested (page 12, lines 22-24 of the revised manuscript).

Second, it might also be helpful to contextualize the doses of sodium nitrate used in vivo. Would similar dosage in humans be tolerable? What was the level of nitrates in the non-supplemented feed? In the average human diet?

Thanks very much for the reviewer 1’s comments. The best dose of sodium nitrate used in the miniature pig in the in vivo study was 2 mmol/kg·day. The coefficient of drug administration from pigs to humans was 0.73; thus, the dose converted to humans was 1.46 mmol/kg·day. For example, for an adult weighing 60 kg, they might have to take 7 g sodium nitrate per day, which is considered a relatively high dose. However, the parotid gland of miniature pigs received 60 Gy irradiation doses in our study, and the period of irradiation lasted for only five days receiving 7.5 Gy each day. When head and neck cancer patients receive irradiation, the irradiation period lasts for two months and they receive only 2 Gy daily, and the total dose would not be given to the parotid gland as most doses are focused on the tumor. In this case, the parotid gland of patients with head and neck cancer could not receive as strong a dose as the experimental animals in this study. Taken together, although the nitrate intake from diet food was 3–7 mg/kg per day for clinical treatment (an adult weighing 60 kg takes 0.18–0.42 g sodium nitrate per day) (Ashworth and Bescos, 2017), it is considered that the dose for clinical treatment could be well tolerated by humans. We have addressed this issue in the Discussion (page 12, lines 1–12) section of the revised manuscript.

Reviewer #2:Following comments are needed to clarify.1. Please check if "Mechanically" in line 26 is right word.

Thanks very much for the reviewer 2’s comments. We have deleted this word from the revised manuscript.

2. Would like to check if nitrate affects free radicals, which are produced during irradiation, and mitochondrial functions. If authors don't know yet, hope authors can try to understand these in their future study.

Thanks very much for the reviewer 2’s comments. Inorganic nitrate can be metabolized into a free radical, nitric oxide (NO), via the nitrate–nitrite–NO pathway, especially under hypoxic or acidic conditions. Irradiation can induce a hypoxic and acidic environment within the salivary glands. Nitrate administration could increase NO levels to exert protective effects of NO.

However, nitrate is a stable substance that cannot be all metabolized to NO under irradiation, while sialin expression increases after nitrate administration, which helps sialin transport more nitrate into cells. It is considered that, except for NO, there are other mechanisms involved in preventing IR damage. It has been revealed that nitrate is related to mitochondrial respiration, activation of key metabolic regulatory pathways, and reduction of oxidative stress (Lundberg, Carlström, Larsen, and Weitzberg, 2011; Lundberg et al., 2018). Thus, more detailed mechanisms, including nitrate regulation of mitochondrial functions, should be the focus of future studies.

We have addressed this issue in the Discussion (page 12, lines 25, 26; page 13, lines 1–5) section of the revised manuscript.